# Egalitarian cooperation linked to central oxytocin levels in communal breeding house mice
Stefan Fischer [1,3,4,5] ✉, Callum Duffield [1,5], William T. Swaney [2], Rhiannon L. Bolton[1], Amanda J. Davidson[1], Jane L. Hurst [1] & Paula Stockley [1] ✉

Relationships between adult females are fundamental to understanding diversity in animal social systems. While cooperative relationships between kin are known to promote fitness benefits, the proximate mechanisms underlying this are not well understood. Here we show that when related female house mice (*Mus musculus domesticus*) cooperate to rear young communally, those with higher endogenous oxytocin levels have more egalitarian and successful cooperative relationships. Sisters with higher oxytocin concentrations in the paraventricular nucleus (PVN) of the hypothalamus weaned significantly more offspring, had lower reproductive skew and spent more equal proportions of time in the nest. By contrast, PVN oxytocin was unrelated to the number of weaned offspring produced in the absence of cooperation, and did not vary in response to manipulation of nest site availability or social cues of outgroup competition. By linking fitness consequences of cooperation with oxytocin, our findings have broad implications for understanding the evolution of egalitarian social relationships.

Explaining variation in cooperative behaviour is a fundamental challenge in evolutionary biology, essential to understanding complex social interactions[1–3]. Among group living vertebrates, relationships between individuals ultimately determine the propensity for cooperative behaviour, varying from close social bonds to tolerance or agonism[4–6]. Female relationships are of particular interest in mammals, since female philopatry is common[7] and provides opportunities for kin selected benefits of cooperation[8,9], potentially favouring the evolution of social bonds and prosocial behaviours[1]. Consistent with this theory, individuals that maintain strong social bonds are known to gain long-term fitness benefits, for example via effects on offspring survival, longevity or shared resource defence[10–13]. However, philopatry can also lead to intense competition among kin for limited resources, potentially resulting in reproductive skew where dominant females monopolise resources or inhibit reproduction of subordinates[4,6,14]. Female relationships are thus fundamental to explaining diversity in mammalian social systems, and have been the subject of much interest from evolutionary and comparative perspectives e.g. refs. 4,15–17. However, to further advance understanding of variation in female relationships ideally requires a combination of both ultimate and proximate perspectives[2,18–21]. A focus on the underlying neural mechanisms is of

particular relevance to understanding patterns of social behaviour and cooperation associated with kin selection.

The neuropeptide oxytocin is known to regulate a range of mammalian social behaviours, particularly in females[21–24]. Oxytocin's nine amino acid peptide sequence has remained highly conserved across vertebrates, and, together with the related peptide arginine vasopressin, has broad relevance to understanding the regulation of social behaviour across diverse animal taxa[25]. Oxytocin is produced primarily in the paraventricular and supraoptic nuclei of the hypothalamus, and has both central actions, through release at different target areas in the brain, and peripheral actions, via release from the posterior pituitary. Beyond established reproductive functions[26], studies of laboratory rodents reveal that centrally released oxytocin modulates a range of social behaviours[23,24,27–30]. In females, these behaviours include social preferences for both opposite and same sex partners[23,30–33]. More broadly, there is evidence in other mammals to suggest that oxytocin may modulate behaviours of both sexes under certain conditions, including studies linking cooperative or affiliative behaviour in natural populations to peripherally measured or administered oxytocin, which is assumed to reflect central actions[13,34–38]. Evidence from laboratory rodents also indicates that prosocial behaviours are targeted to preferred partners through reinforcement by

[1]Mammalian Behaviour & Evolution Group, Department of Evolution, Ecology and Behaviour, University of Liverpool, Leahurst Campus, Neston, CH64 7TE, UK. [2]School of Biological and Environmental Sciences, Liverpool John Moores University, Byrom Street, Liverpool, L3 3AF, UK. [3]Present address: Konrad Lorenz Institute of Ethology, Department of Interdisciplinary Life Sciences, University of Veterinary Medicine Vienna, Savoyenstrasse 1, 1160 Vienna, Austria. [4]Present address: Department of Behavioral & Cognitive Biology, University of Vienna, University Biology Building (UBB), Djerassiplatz 1, 1030 Vienna, Austria. [5]These authors contributed equally: Stefan Fischer, Callum Duffield. ✉e-mail: Stefan.Fischer@vetmeduni.ac.at; p.stockley@liverpool.ac.uk

reward and social memory systems known to be modulated by forebrain oxytocin signalling[39]. Oxytocin may therefore facilitate the maintenance of social bonds that result from cooperative and other prosocial behaviours between female kin in group-living mammals. However, the extent to which oxytocin explains variation in cooperative female relationships, or associated fitness consequences of cooperation, remains unexplored.

House mice (*Mus musculus domesticus*) are an ideal model species in which to test these questions. Female house mice are facultative communal breeders, often choosing to combine their offspring in a shared nest and cooperating to nurse the resulting communal litter indiscriminately[40]. However, since lactation is energetically costly, communal nursing involves a risk of exploitation, with potential for unequal investment in the shared litter[41]. The choice of a communal nesting partner is therefore important to individual reproductive success[11], and females typically prefer close relatives[42,43]. Nonetheless, even among related females, cooperative relationships can vary from relatively egalitarian to despotic, where one female gains fitness benefits at the expense of relatively greater investment by their partner, and may benefit by inhibiting their partner's reproductive success[40]. If oxytocin modulates cooperative behaviours, it may therefore be associated with the decision by female house mice to nest communally with a preferred partner, or with relatively successful or egalitarian cooperative relationships.

House mice are also a useful model species to test how oxytocin signalling might vary in response to contrasting long-term environmental conditions. Social units typically consist of a dominant male, several breeding females and their offspring[44]. A commensal existence with humans means that food is not usually limiting for wild house mice under natural conditions, as typically they live in close proximity to abundant food supplies in agricultural or domestic settings. Rather, females compete for safe nest sites needed for successful reproduction within shared territories[45]. Competition between female kin groups can occur both within and between social units, and competition for the safe nest sites needed for successful reproduction is particularly intense at high population density[45,46]. Hence, if plasticity exists in central oxytocin production[47], elevated levels might facilitate increased social tolerance when resources such as safe nest sites are limited[48,49]. Further, consistent with reports of increased oxytocin release in response to outgroup competition in other species, persistent competition with unrelated females might lead to increased oxytocin production in female house mice, potentially promoting cooperation between kin to defend key resources[25] or alleviate stress[50].

To test these predictions, we manipulated the social environment of related female house mice breeding in enclosures under carefully controlled conditions. Specifically, we manipulated the availability of protected nest sites preferred for breeding, the relatedness of competitors living in the same territory, and the presence or absence of neighbours living in an adjacent territory. These manipulations each represent realistic scenarios for commensally living wild house mice[45,46]. Previous studies in wild animals have quantified short-term effects of peripheral oxytocin administration on cooperative behaviours[34,37], or associations between cooperative behaviours and peripheral oxytocin levels[35,36]. Our complementary approach using wild-derived house mice instead explores longer-term causes and consequences of variation in central oxytocin, measured as basal concentration in the paraventricular nucleus (PVN) of the hypothalamus, the main source of centrally-released oxytocin[51,52]. We conducted two experiments (Fig. 1) to investigate if cooperative behaviour and reproductive success of sister dyads is related to variation in their oxytocin production, and if oxytocin levels in the PVN are influenced by: i) limited availability of protected nest sites, or ii) long-term exposure to social cues of outgroup competitors, either within the same territory or as neighbours in an adjacent territory. Experiments allowed sufficient time for social groups to become established, for subjects to complete a reproductive cycle, and for a post-breeding recovery period prior to quantifying oxytocin levels. PVN oxytocin concentrations were quantified at the end of each experiment to test for predicted relationships between basal oxytocin levels and cooperative relationships, and for differences in basal oxytocin levels explained by experimental conditions.

## Results

Each experiment consisted of a series of independent trials (16 for experiment 1 and 20 for experiment 2 – see Supplementary Table 1), in which the environment of female house mice was manipulated (Fig. 1). Subjects within each trial were a pair of littermate sisters living with two younger non-breeding females. This design replicates a naturally occurring social structure, where older females tend to monopolise breeding opportunities[45,46]. Experiment 1 manipulated the availability of protected nest sites and the relatedness of younger competitors living within the subjects' territory, while experiment 2 manipulated the presence or absence of unrelated competitors in a neighbouring territory (Fig. 1). The experiments thus included subjects experiencing outgroup competition with unrelated females either within their own territory or in a neighbouring territory.

PVN oxytocin concentrations were quantified for 64 subjects ($n = 24$ in experiment 1 and $n = 40$ in experiment 2) from 34 trials. These include cases where PVN oxytocin concentrations were quantified for both subjects in 30 trials, and for one subject in four trials. At least one subject within a sister dyad produced weaned offspring in 33 trials, and maternity was assigned unambiguously for 252 of 256 weaned offspring produced by subjects with known PVN oxytocin concentrations. This resulted in full maternity allocation of weaned offspring for 28 sister dyads, including 15 cases where a communal nest was formed and 13 where no communal nest was formed and only one subject produced weaned offspring. Where both subjects in a trial produced a litter, they always chose to combine their offspring within a protected nest box, even in cases where more than one protected nest box was available (see Supplementary Table 1).

### Variation in PVN oxytocin concentrations

PVN oxytocin concentrations varied from 162.9 to 1733.6 pg/mg protein (mean $\pm$ SE = 732.2 $\pm$ 45) and were significantly correlated within sister dyads (Supplementary Table 2 factor: 'Individual PVN oxytocin' [$F_{1,24.79} = 21.16$], $p < 0.01$); Fig. 2). However, variation in individual PVN oxytocin concentrations was not explained by subjects' contrasting experience of the number of protected nest sites available (Table 1 factor: 'Protected nest sites' [$F_{1,30.31} = 6e^{-4}$, $p = 0.98$]), or by contrasting experience of outgroup competition with unrelated females in their own or a neighbouring territory (Table 1 factor 'Outgroup competition' [$F_{2,27} = 1.1$, $p = 0.35$]). Similarly, PVN oxytocin concentrations were not significantly influenced by subjects' age or body mass (Supplementary Table 3), whether they successfully reared offspring or not (Supplementary Table 4 factor 'Whether subject produced weaned offspring' [$F_{1,47.12} = 0.03$, $p = 0.85$]), or by the time between removal of weaned offspring and collection of PVN samples (Supplementary Table 5 factor 'Number of days' [$F_{1,24.57} = 1.08$, $p = 0.31$]).

### PVN oxytocin concentrations and reproductive success

Average PVN oxytocin concentrations of sister dyads did not predict whether or not they both bred and formed a communal nest (Supplementary Table 6 factor 'Average PVN oxytocin concentration [$\chi^2 = 0.72$, $p = 0.39$]). However, when testing if oxytocin explained variation in the combined reproductive success of sisters within dyads, we found a significant interaction between average PVN oxytocin concentration and whether or not offspring were reared communally (Table 2a factor 'Average PVN oxytocin x Communal nest' [$F_{1,23} = 5.92$, $p = 0.02$], Fig. 3). Further analysis confirmed that subjects' average PVN oxytocin concentrations were positively related to the total number of weaned offspring produced when a communal nest was formed, but not when only one subject bred successfully (Table 2b, c). To further explore if PVN oxytocin predicted reproductive success independently of cooperative behaviour, we also tested if PVN oxytocin concentrations explained significant variation in the number of weaned offspring produced by individual breeding subjects. This might be expected if oxytocin levels predict the quality of maternal care provided, independent of the relationship between sister dyads. However, we found no evidence that individual PVN oxytocin concentrations explained variation in subjects' weaned offspring numbers (Supplementary Table 7 factor 'Individual PVN oxytocin concentration' [$F_{1,40} = 0.01$,

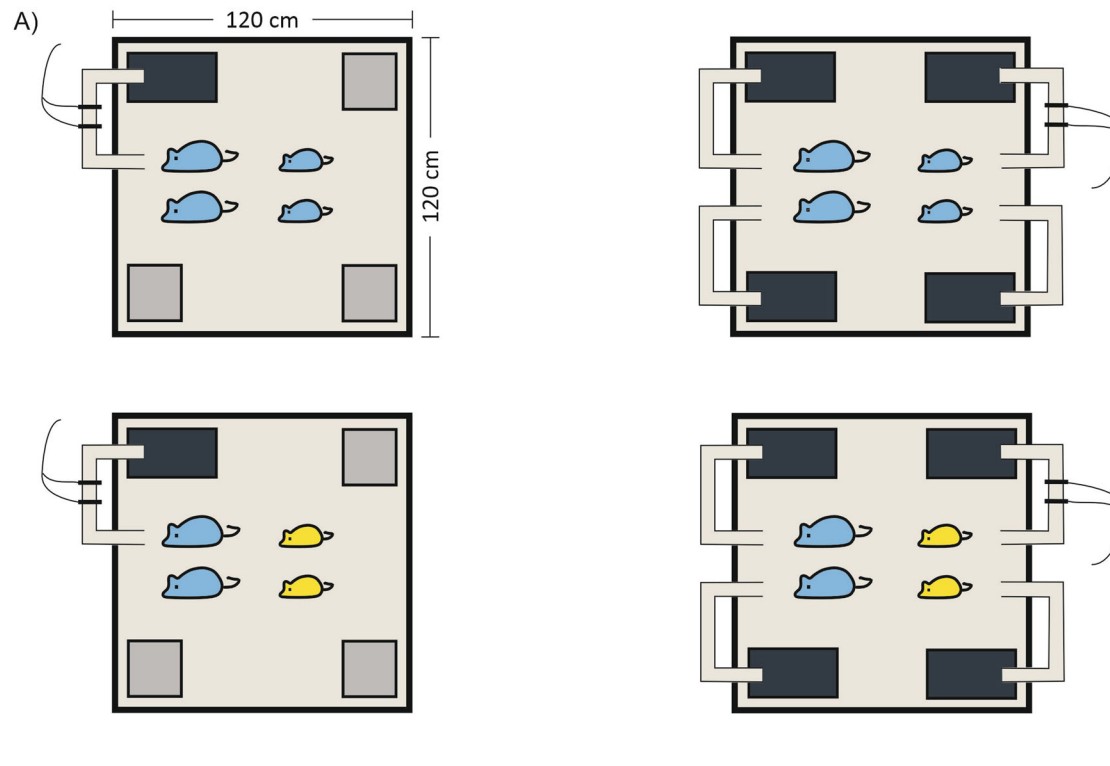

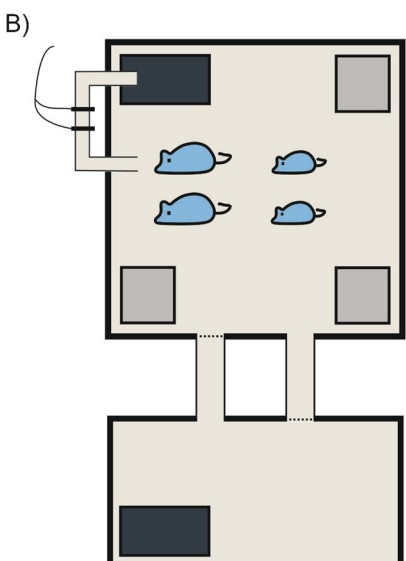

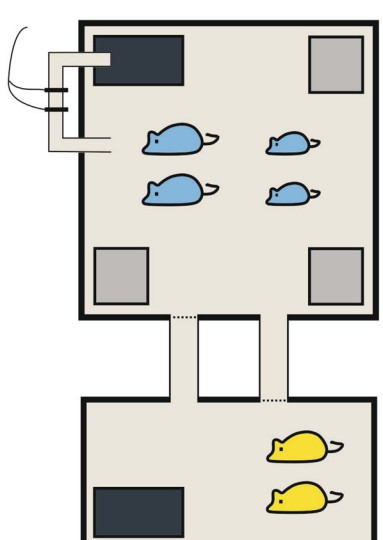

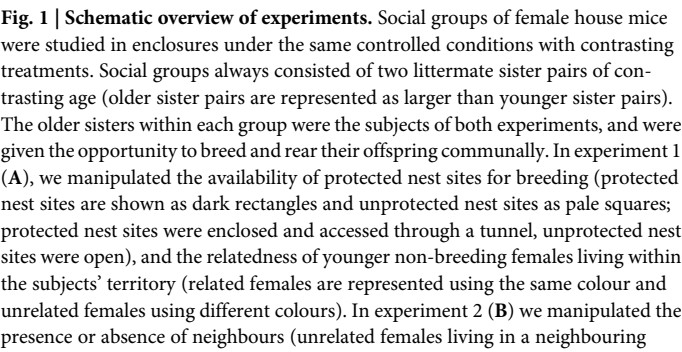

**Fig. 1 | Schematic overview of experiments.** Social groups of female house mice were studied in enclosures under the same controlled conditions with contrasting treatments. Social groups always consisted of two littermate sister pairs of contrasting age (older sister pairs are represented as larger than younger sister pairs). The older sisters within each group were the subjects of both experiments, and were given the opportunity to breed and rear their offspring communally. In experiment 1 (**A**), we manipulated the availability of protected nest sites for breeding (protected nest sites are shown as dark rectangles and unprotected nest sites as pale squares; protected nest sites were enclosed and accessed through a tunnel, unprotected nest sites were open), and the relatedness of younger non-breeding females living within the subjects' territory (related females are represented using the same colour and unrelated females using different colours). In experiment 2 (**B**) we manipulated the presence or absence of neighbours (unrelated females living in a neighbouring territory), linked by connecting tunnels blocked with wire mesh (shown as dotted lines). Each enclosure contained two transponder readers that monitored the nest site attendance of subjects in occupied protected nest sites during post-natal day 0–14 (shown on the tunnel entrance to a protected nest site within each enclosure). **A** Upper left: older and younger sister pairs are related, and a single protected nest site is available. Upper right: older and younger sister pairs are related, and four protected nest sites are available. Lower left: older and younger sister pairs are unrelated, and a single protected nest site is available. Lower right: older and younger sister pairs are unrelated, and multiple protected nest sites are available. **B** Left: older and younger sister pairs are related, a single protected nest site is available, and no neighbours are present in an adjacent territory. Right: older and younger sister pairs are related, a single protected nest site is available, and neighbours are present in an adjacent territory.

$p = 0.91$]). By contrast to our analysis of average PVN oxytocin concentrations of sister dyads, this was irrespective of whether subjects formed a communal nest or not, as shown by a non-significant interaction term 'Individual PVN oxytocin concentration x Communal nest' ($F_{1,29} = 1.9$, $p = 0.18$), which was subsequently dropped from the final model. We therefore tested if average PVN oxytocin levels of communally breeding subjects predicted the degree of reproductive skew between them (difference in number of weaned offspring), controlling for total offspring numbers, since this might explain why PVN oxytocin concentrations explained variation in combined but not individual reproductive success. Consistent with this interpretation, the analysis revealed that communally breeding subjects with higher average oxytocin concentrations produced more similar numbers of weaned offspring after controlling for combined litter size, whereas there was greater asymmetry in the number of offspring weaned by subjects with lower average oxytocin concentrations (Supplementary Table 8, factor 'Average PVN oxytocin' [$F_{1,10} = 5.42$, $p = 0.04$]).

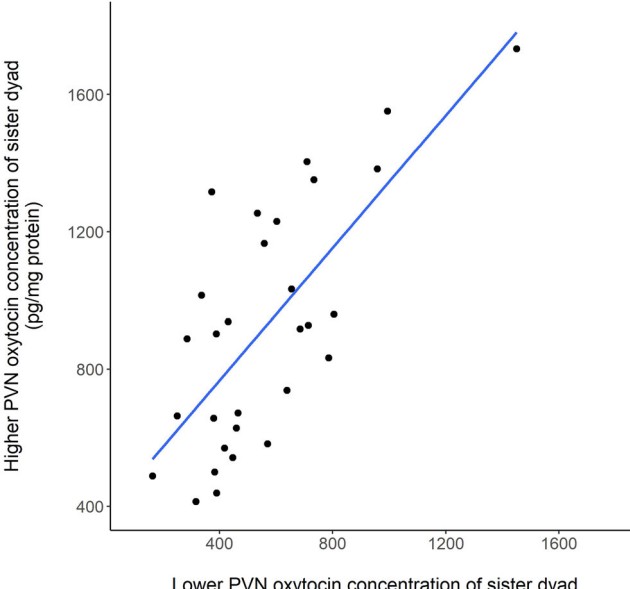

**Fig. 2 | Relationship between PVN oxytocin concentrations of sister dyads from the same social group.** Subjects were from two experiments with independent manipulation of access to protected nest sites and outgroup competition. Subjects within each trial were classified as having a higher or lower PVN oxytocin concentration relative to one another. PVN oxytocin concentrations of sister dyads within the same social group were significantly correlated, irrespective of the experimental treatments (protected nest site availability and outgroup competition). See Supplementary Table 2 for statistical analysis.

## PVN oxytocin concentrations and cooperation in the communal nest

We quantified relative time spent in the communal nest as a measure of cooperative behaviour when both subjects reared offspring together. As expected, an assay of feeding behaviour confirmed that time spent in the nest was negatively correlated with time spent feeding (Supplementary Table 9 factor 'Time spent feeding' [$F_{1,18.5} = 9.61$, $p = 0.01$], Supplementary Fig. S1), and hence to individual lactational investment[53] (see Methods for further explanation).

Average PVN oxytocin concentrations of sister dyads were significantly related to the relative time they spent in the nest from birth to postnatal day 14 during active (dark) periods (Table 3 factor: 'Average PVN oxytocin' [$F_{1,13} = 7.34$, $p = 0.02$], Fig. 4). That is, sisters with higher average oxytocin levels spent more similar proportions of time in the communal nest, whereas there was greater asymmetry in time in the nest between subjects with lower average oxytocin levels. Outgroup competition with unrelated females in the subjects' own or a neighbouring territory and the availability of protected nest sites (multiple or single) had no significant influence on relative time spent in the communal nest and were dropped from the final model (Table 3). Average oxytocin levels did not predict relative time in the nest during inactive (light) periods (Supplementary Table 10 factor: "Average PVN oxytocin" [$F_{1,8} = 0.16$, $p = 0.7$], although a significantly greater skew in time spent in the nest during inactive (light) periods was associated with the presence of outgroup competitors in the neighbouring territory (Supplementary Tables 10, 11). Finally, PVN oxytocin concentrations of individual subjects were not associated with the absolute duration of time they spent in the nest, during either active (dark) or inactive (light) periods (Supplementary Table 12).

## Discussion

We found that sister dyads with higher PVN oxytocin concentrations had higher combined reproductive success and lower reproductive skew when cooperating to rear offspring communally. By contrast, PVN oxytocin concentrations did not explain variation in the reproductive success of females breeding alone. PVN oxytocin concentrations of sister dyads sharing offspring care also predicted the relative time they spent in the communal nest during active periods, as an indirect measure of lactational investment in the shared litter. That is, the proportion of time spent in the nest was more similar between sisters that had higher average PVN oxytocin concentrations. Collectively, these results suggest that oxytocin may have fitness consequences through its effects on female social relationships, with higher levels mediating more successful and egalitarian cooperation between sisters when breeding communally. Thus, although previous studies suggest that laboratory mouse strains are unsuitable subjects for studying the role of oxytocin in social relationships[54,55], our findings indicate that wild-derived house mice offer a useful model system, particularly in the context of communal breeding. Wild-derived house mice are more suitable subjects than laboratory strains for studying social relationships because

## Table 1 | No effect of protected nest site availability or outgroup competition on individual PVN oxytocin concentrations

| Factors | Estimate ± SE | Num. D.F. | Den. D.F | F-value | P-value |
|---|---|---|---|---|---|
| Intercept | 6.54 ± 0.2 | – | – | – | – |
| Protected nest sites (*multiple, single*) | $-4e^{-3} ± 0.2$ | 1 | 30.31 | $6e^{-4}$ | 0.98 |
| Outgroup competition(*yes[same territory] / yes[neighbouring territory] / no*) | – | 2 | 27 | 1.1 | 0.35 |
| Outgroup competition (*yes[same territory]*) | $-0.29 ± 0.2$ | – | – | – | – |
| Outgroup competition (*yes[neighbouring territory]*) | $-0.05 ± 0.17$ | – | – | – | – |

Subjects were from two experiments with independent manipulation of access to protected nest sites (factor 'Protected nest sites') and outgroup competition (factor 'Outgroup competition'). Results are shown from a linear mixed model. The levels of each factor are shown in parenthesis after the factor name: Protected nest sites: '*multiple*': multiple protected nest sites; '*single*': single protected nest site. Outgroup competition: '*yes[same territory]*': unrelated competitors were present in the same territory; '*yes[neighbouring territory]*': unrelated competitors were present in a neighbouring territory; '*no*': no outgroup competition. Estimates are shown on a log scale and as differences to the reference levels 'multiple' for the factor 'Protected nest sites' and 'no' for the factor 'Outgroup competition'. Experiment (1 or 2), age and body mass did not predict the PVN oxytocin concentrations of subjects and were removed from the final model. To obtain normally distributed residuals the dependent variable was log transformed. To obtain p-values an F-test was used to compare models with and without the factor of interest. N = 64 females in 34 trials across 10 blocks and two experiments (Supplementary Table 1).

**Table 2 | Factors predicting the combined number of weaned offspring produced by sister dyads**

| Factors | Estimate ± SE | Num. D.F. | Den. D.F | F-value | *P*-value |
|---|---|---|---|---|---|
| **(a) All trials** | | | | | |
| Intercept | 7.45 ± 2.05 | – | – | – | – |
| Average PVN oxytocin (pg/mg protein) | $-4e^{-3} \pm 3e^{-3}$ | 1 | 23 | $2e^{-3}$ | 0.97 |
| Communal nest (yes, no) | 0.45 ± 2.68 | 1 | 23 | 0.03 | 0.87 |
| Average PVN oxytocin x Communal nest | $0.01 \pm 3e^{-3}$ | 1 | 23 | 5.92 | **0.02** |
| **(b) Trials where both females produced weaned offspring and shared a communal nest** | | | | | |
| Intercept | 7.91 ± 1.53 | – | – | – | – |
| Average PVN oxytocin (pg/mg protein) | $4e^{-3} \pm 2e^{-3}$ | 1 | 12 | 5.15 | **0.04** |
| **(c) Trials where only one female produced weaned offspring** | | | | | |
| Intercept | 7.45 ± 2.28 | – | – | – | – |
| Average PVN oxytocin (pg/mg protein) | $-4e^{-3} \pm 3e^{-3}$ | 1 | 11 | 1.92 | 0.19 |

Subjects were from two experiments with independent manipulation of access to protected nest sites and outgroup competition. Results are shown from three separate linear mixed models for (a) all trials, (b) trials where both subjects produced weaned offspring and shared a communal nest, and (c) trials where only one subject produced weaned offspring. Experimental treatments (protected nest site availability and outgroup competition), subjects' age, body mass and experiment (1 or 2) did not influence the combined number of weaned offspring produced by sister dyads and were dropped from the final model in (a). The estimate in (a) is shown as difference to the reference level 'no' for factor 'Communal nest' (whether a communal nest was formed). To obtain *p*-values F-tests were used to compare models with and without the factor of interest. Values in bolded text are statistically significant ($P < 0.05$). (a) $N = 27$ sister pairs across 10 blocks and two experiments; (b) $N = 14$ sister pairs across eight blocks and two experiments; (c) $N = 13$ sister dyads across eight blocks and two experiments (Supplementary Table 1).

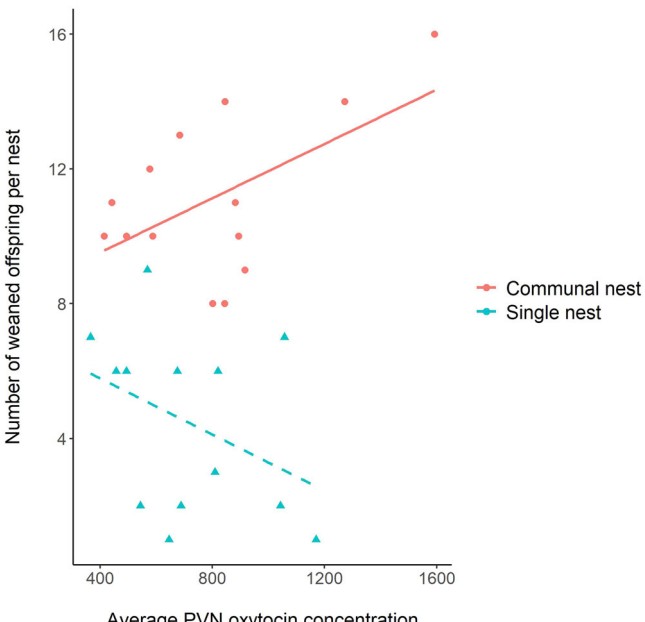

**Fig. 3 | Relationship between the total numbers of offspring weaned per nest and the average PVN oxytocin concentration of sister dyads, according to whether or not the sister dyads reared young communally.** Subjects were from two experiments with independent manipulation of access to protected nest sites and outgroup competition. Higher average PVN oxytocin concentration of sister dyads was associated with a greater total number of weaned offspring when females cooperated to rear offspring in a communal nest ('Communal nest', red, solid line) but not when offspring were reared by a single female ('Single nest', blue, dashed line). See Table 2 for statistical analysis.

**Table 3 | Factors predicting relative time spent in the nest by sister dyads cooperating in a communal nest during the active (dark) period**

| Factors | Estimate ± SE | Num. D.F. | Den. D.F | F-value | *P*-value |
|---|---|---|---|---|---|
| Intercept | 0.16 ± 0.03 | – | – | – | – |
| Average PVN oxytocin (pg/mg protein) | $-1e^{-4} \pm 3e^{-5}$ | 1 | 13 | 7.34 | **0.02** |

Subjects were from two experiments with independent manipulation of access to protected nest sites and outgroup competition. Results are shown from a linear mixed model. Relative time spent in the nest was calculated as the difference in the proportion of total time spent in the nest by each subject. Experimental treatments (protected nest site availability and outgroup competition) and experiment (1 or 2) did not influence relative time spent in the nest and were dropped from the final model. To obtain *p*-values an F-test was used to compare models with and without the factor of interest. Values in bolded text are statistically significant ($P < 0.05$). $N = 15$ sister dyads across eight blocks and two experiments.

laboratory strains do not have the normal variation in individual genetic identity signals that are essential for individual and kin recognition[56], and because the competitive responses of laboratory strains have been greatly reduced through artificial selection.

Many social mammals show consistent preferences for particular individuals, often described as social bonds[57], and there is growing evidence for the adaptive value of such relationships, beyond established functions in mating and parental care[10–13]. Previous studies demonstrate that oxytocin is involved in mediating social bonds in a variety of contexts[20,21,39,58]. Although oxytocin has not been demonstrated to influence social bonds between female house mice[49,54], there is evidence that adult females form consistent social preferences prior to breeding communally, with consequences for reproductive success[11,48]. Females paired with a preferred partner are more likely to establish an egalitarian cooperative relationship, resulting in higher reproductive success compared to those with non-preferred partners[11]. Variation in reproductive success is at least partly explained by differences in the early mortality of young, which is minimised when females breed with a familiar close relative[40]. In the context of the present study, it appears that relationships between sisters with relatively high oxytocin levels may involve a lesser degree of social tension, resulting in relatively low reproductive skew. Conversely, females with relatively low oxytocin levels may experience a greater degree of social tension, potentially leading to an increased risk of infanticide[40] or inhibition of fertility[59,60], which in turn could result in lower combined reproductive success and greater reproductive skew.

We found no evidence that PVN oxytocin concentrations explained variation in reproductive success of females breeding alone, or of individual females. Absolute time spent in the nest with pups was also unrelated to PVN oxytocin levels. Hence, we found no evidence that relatively high PVN oxytocin levels were associated with more or better-quality maternal care per se. Although studies of laboratory mice confirm that oxytocin has an essential role in milk ejection, such studies also report normal levels of maternal behaviour in oxytocin deficient females[61–63], including those with conditional knock-out of oxytocin in the PVN[64]. Hence our finding that

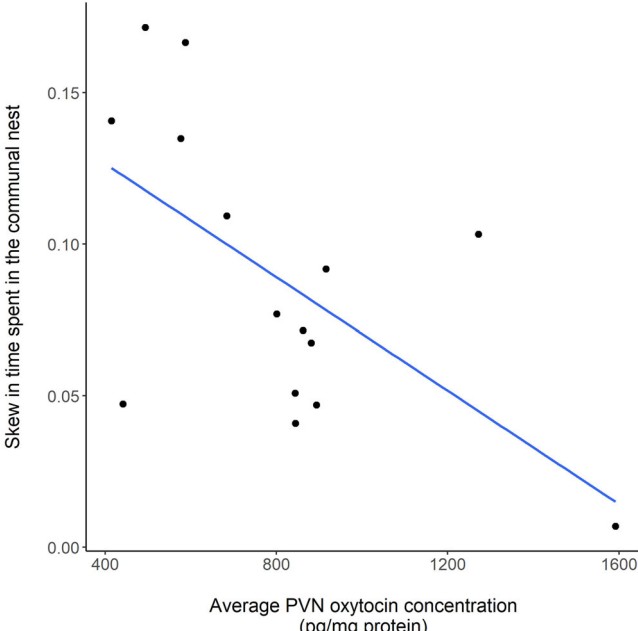

**Fig. 4 | Relationship between skew in time spent in the communal nest by communally breeding sister dyads during the active (dark) period and their average PVN oxytocin concentrations.** Subjects were from two experiments with independent manipulation of access to protected nest sites and outgroup competition. To analyse the relative time spent in the nest by communally breeding sister dyads we calculated the difference in the proportion of total time spent in the nest by each subject. Higher average PVN oxytocin concentration of sister dyads was associated with a lower skew in the relative time they spent in the communal nest, indicating that each subject spent more similar time in the nest around the time of peak maternal investment. See Table 3 for statistical analysis.

natural variation in PVN oxytocin levels of wild-derived house mice has no apparent influence on their weaned offspring numbers or maternal care behaviour in the absence of communal breeding is not unexpected. Rather, our findings suggest that the PVN oxytocin levels of sister dyads reflect variation in the strength of their social relationships, with consequences for the combined reproductive success and reproductive skew of sisters that cooperate to breed communally. Also consistent with oxytocin-mediated social relationships between females, we found evidence that cooperation appears to be more egalitarian between sister dyads with higher average PVN oxytocin concentrations, as they spent more similar proportions of time in the communal nest during active periods. Although sister dyads with relatively high oxytocin levels also contributed more similar numbers of offspring to the communal litter, this is unlikely to explain why they spent more similar time in the nest. It is well established that female house mice nurse offspring in communal nests indiscriminately, investing in the total litter regardless of the proportion or number of their own offspring in the nest[41,43,53,65,66]. More equal investment in time spent with offspring or time spent foraging by sisters with higher PVN oxytocin levels might instead reflect active coordination in providing offspring care, as a strategy for minimising conflict[67,68].

No direct causal link between oxytocin and behaviour is demonstrated by our results, and we are unable to confirm directionality in reported relationships between oxytocin levels and behaviour. Nonetheless, by focusing on relationships with central oxytocin levels in the PVN, our study avoids common difficulties of interpreting measures or manipulations of peripheral oxytocin levels, which may be unrelated to central actions. Moreover, unlike most previous studies, we focus on quantifying basal PVN oxytocin levels rather than behaviour-associated oxytocin release. Relationships between basal PVN oxytocin expression and social behaviour have been demonstrated previously using rodent models[27], and in the context of

communal breeding, basal oxytocin levels may also predict oxytocin release in response to social interactions with the cooperating partner. Since our aim was to allow subjects to exhibit natural behaviour over a complete breeding cycle, we did not attempt to quantify central oxytocin release during periods of cooperation, when lactation may also have influenced our findings. However, our results suggest that investigating the relationship between basal and activated PVN oxytocin expression may be a promising avenue for further investigation into the proximate mechanisms that regulate social behaviours. Moreover, the approach used here does not allow for differentiation between the PVN's functionally distinct magnocellular and parvocellular neuronal populations[69], further exploration of which might offer additional insights in future studies.

Correlated oxytocin production within sister dyads could be a result of shared genetic, environmental or social factors. For example, there is evidence for plasticity in PVN development linked to early life social experience[70], and in hypothalamic oxytocin expression linked to environmental stressors[71]. PVN oxytocin expression also shows plasticity in adult animals, at least under artificially stressful conditions[72,73]. However, we found no evidence here of plasticity in basal PVN expression under more naturalistic conditions, such as whether or not subjects had limited access to protected nest sites, or experienced outgroup competition with unrelated conspecifics. Combined with the similarity in sisters' oxytocin levels, this might suggest that basal oxytocin expression is a heritable trait that shows relatively little plasticity in adult animals under normal conditions, perhaps reflecting variation in the number or size of oxytocin neurons[28,51,52,74]. However, plasticity in hypothalamic oxytocin neural densities can occur in certain social contexts[47], and future studies employing immunohistochemistry to characterise variation in numbers and types of oxytocin neurons in the PVN could provide further useful insights. Alternatively, oxytocin receptor binding may respond more flexibly to environmental conditions than the basal oxytocin expression levels measured here[23,75]. However, if oxytocin production is influenced by social feedback mechanisms[35], then correlated levels between sisters could also be causally linked to the strength of their social relationship[76]. Consistent with this idea, previous evidence suggests that female house mice are more likely to show egalitarian cooperation if they form an affiliative social relationship prior to breeding[11]. Although such relationships are more usually formed between kin, both affiliative social relationships and egalitarian cooperation can also occur between unrelated females[11,53]. Hence there is potential for future studies to tease apart the effects of relatedness and strength of social relationships in explaining the association between hypothalamic oxytocin levels and egalitarian cooperation reported here.

Despite growing evidence linking oxytocin to outgroup responses[13,25], we found no evidence of plasticity in oxytocin production linked to outgroup competition. Nonetheless, it is unlikely that subjects in our study were unresponsive to competitors. For example, we found that the presence of neighbours influenced relative time spent in the nest during periods when the mice are typically resting, and previous studies demonstrate that female house mice respond to social cues of potential competitors through elevated investment in scent marking, linked to territorial defence[77–79]. Oxytocin may not therefore be a significant factor in the response of female house mice to outgroup competition. Alternatively, plasticity in basal oxytocin expression in response to out-group competition may be dependent on aspects of subjects' phenotype that were not quantified in the current study. For example, responsiveness to an out-group threat might vary according to a subject's competitive behaviour or dominance status[80]. Moreover, although our study was designed to imitate natural conditions, subjects were not free-living and, for ethical reasons, we managed social interactions between unrelated competitors to avoid a risk of escalated aggression. Hence, although subjects were continuously exposed to social odours of competitors, reinforced with controlled physical contact, we cannot rule out that their response to competitors may be different in unconstrained natural populations.

Oxytocin has previously been associated with cooperative behaviour in wild mammals, including examples of alloparental care[34], social grooming[35], food sharing[36], and cooperative defence[13]. Here, although we found that

sister dyads with higher average oxytocin levels had more egalitarian cooperative relationships when breeding communally, we found no evidence that higher oxytocin levels were associated with communal breeding per se. This is consistent with evidence that oxytocin does not act unilaterally to increase prosocial behaviour but rather functions in a context specific way by modulating attention to relevant social cues[29]. Oxytocin may therefore influence degrees of cooperation with a given partner rather than the propensity to cooperate. Similarly, in a study of food sharing in vampire bats, Carter and Wilkinson[37] found peripheral oxytocin treatment did not affect the probability for cooperation between dyads, but did lead to a greater degree of cooperation via an increase in the size of food donations.

In conclusion, our study links the fitness consequences of cooperative relationships between female kin to variation in hypothalamic oxytocin levels. This suggests a role for oxytocin in mediating kin selected benefits of cooperation, with implications for explaining the evolution of egalitarian social relationships. Social competition and conflict shape the social systems of group living animals, with diverse outcomes influenced by kin selection and benefits of cooperation[81,82]. The resulting tension between competition and cooperation is reflected by variation in how benefits of cooperative behaviour are distributed between group members. In egalitarian social systems, benefits are shared relatively evenly according to effort invested, whereas in despotic social systems, benefits are more likely to accrue disproportionately to dominant individuals at the expense of others[4,83]. Our study provides evidence of variation in the balance between egalitarian and despotic outcomes linked to central oxytocin levels of cooperating individuals. If similar variation is replicated across species, this could help us to understand the proximate factors influencing egalitarian and despotic social behaviours, hence providing broad insight into social system diversity. Our study thus offers potential new insights for understanding both the proximate basis of cooperative behaviour and the evolution of diverse social systems.

## Methods
### Subjects and husbandry
House mice used in this study were from a captive outbred colony, derived from wild ancestors originating from several populations in the northwest of England, UK, with regular addition of new wild-caught animals. The colony is maintained under controlled environmental conditions (temperature 20–21 °C, relative humidity 45–65%, and a reversed 12:12 h light cycle with lights off at 08:00). All animals are provided with *ad libitum* access to water and food (Lab Diet 5LF2 Certified Rodent Diet, Purina Mills, USA), and housed on Corn Cob Absorb 10/14 substrate with paper wool nest material. Subjects were bred in standard laboratory cages (MB1, North Kent Plastics, UK; $45 \times 28 \times 13$ cm) with behavioural enrichment and use of handling tunnels to minimise stress[84], and were nulliparous at the start of the experiment. Radio frequency identification (RFID) tags injected under the nape skin were used for individual identification of adult mice. Unrelated animals were classed as those with no shared full sibling grand-parents (r < 0.032).

### Experimental design
We manipulated social competition within and between female kin groups in two experiments (Fig. 1). Social groups in each experiment consisted of two littermate sisters of contrasting age. The older sisters were the subjects of both experiments, and were given the opportunity to breed and to rear their offspring communally. Both experiments were conducted in blocks under the same controlled conditions (see Supplementary Note 1). In experiment 1, the environment was manipulated by varying the availability of protected nest sites, and the relatedness of younger females living within the subjects' territory (two sisters or an unrelated sister dyad of equivalent age)[77]. In experiment 2, the environment was manipulated by varying the presence or absence of neighbours in an adjacent territory, and by regularly exposing subjects to social odours of these neighbours, including territorial intrusions, or a control treatment without neighbours. In both experiments, younger females were transferred to an MB1 cage inside the enclosure, to prevent mating and infanticide, when a breeding male was introduced. During this

period, younger females were released regularly to maintain their social odours throughout the enclosure (see Supplementary Note 1). Time spent with offspring by subjects was automatically monitored using transponder readers connected to a customized data logger (Francis Scientific Instruments, UK). Initial sample sizes were $n = 32$ subjects (16 sister dyads) for experiment 1 and $n = 40$ subjects (20 sister dyads) for experiment 2. An overview of sample sizes subsequently available for analysis in each component of the study can be found in Supplementary Table 1.

### Quantifying variation in cooperative behaviour
To determine if subjects bred communally, we combined behavioural observations of nest sharing with maternity analysis of weaned offspring to confirm that both subjects had contributed to a communal litter (see Supplementary Note 1). For subjects sharing offspring care, we also quantified the relative time each spent in the communal nest as a measure of cooperative behaviour, assuming that subjects spending more similar amounts of time in (or out of) the nest were showing more egalitarian cooperation. We quantified time spent in the nest for the active (dark: 08.00-20.00) and the inactive (light: 20.00-08.00) periods separately, as mice spend significantly more time resting in the nest during the light period. Nests were checked daily for pups and recording was initiated on detection of the first litter born within an enclosure. Time spent in the nest was then recorded continuously for 15 days, reaching the period of peak lactation at around postnatal day 14. Recordings were subsequently analysed blind to treatment group to give the total time spent by each subject in the nest.

Counter-intuitively, time spent in the nest has previously been shown to correlate negatively with maternal lactational investment and offspring growth, due to a trade-off with time spent foraging to produce milk[53]. To confirm that time spent in the nest by cooperating females is a useful indirect measure of their relative investment in the communal litter, we therefore assayed time spent feeding during lactation for a subset ($n = 21$) of subjects in experiment 2. This assay was conducted between postnatal days 10–14, to confirm that time spent in the communal nest trades-off against time spent feeding by lactating females[53]. Behaviour was recorded using overhead cameras over four consecutive days. The duration of feeding was then quantified for each subject from video recordings during the active (dark) period between 09.00-10.00, 12.00-13.00 and 19.00-20.00 each day, using RFID tag data matched with video recordings to identify individuals.

### Quantifying variation in central oxytocin levels
Offspring were removed at age 28–30 days, approximately 5–7 days after the cessation of lactation[85]. After weaned offspring were removed, subjects were kept within their experimental treatments for a minimum of one week prior to measurement of PVN oxytocin levels (median 16 days, range 8–34 days). For each block, all subjects were killed humanely on the same day, and variation in the timing of sample collection was taken into account in statistical analyses by including block as a random effect (see Methods). We confirmed that the time between removal of weaned offspring and collection of PVN samples had no significant effect on measured oxytocin concentrations (Supplementary Table 5). We also compared findings for subjects that had produced offspring versus those that had not, further confirming that recent lactation had no influence on PVN oxytocin levels (Supplementary Table 4).

Whole brains were removed within 5 min, frozen in hexane on dry ice, and transferred to storage at −80 °C. Brain micro-dissections were carried out using a Leica Cryostat at −20 °C with stereotaxic coordinates for reference[86]. Brains were centrally mounted using optimal cutting temperature (OCT) embedding medium (Solmedia, UK). The PVN of the hypothalamus was removed at Bregma −0.34 mm using a 1 mm biopsy punch (Selles Medical, UK), placed into a frozen 1.5 ml microcentrifuge tube and quickly transferred to dry ice before storage at −80 °C.

Oxytocin was analysed in PVN homogenate using a commercially available EIA (ADI-900-153, Enzo, USA) following manufacturer's instructions. Validations of parallelism and accuracy were conducted satisfactorily (see Supplementary Note 1). All samples were run in duplicate alongside a

standard curve (in triplicate) on a total of five plates. In addition to the in-plate controls supplied by the manufacturer, two further controls were used in duplicate on each plate at 25% (control 1) and 75% (control 2) binding. Intra-assay CVs were less than 10% for all samples included in the analyses (average 2.6%), and inter-assay CVs were less than 15% (average 13.4%).

As each 1 mm micro-punch was assumed to contain slight variation in the precise weight of PVN tissue sampled, oxytocin was normalised to protein concentration within the final eluted PVN homogenate (see Supplementary Note 1). Oxytocin levels in tissue extracts could then be expressed as pg/mg total protein.

### Statistical analysis

For statistical analysis we used R 4.1.1[87], with the package lme4[88]. All statistical tests are two-tailed. We used linear mixed models (LMM), and one generalized linear mixed effect model (GLMM). To obtain p-values, we used the drop1() function to perform a likelihood ratio test (F-test for LMMs, $\chi^2$-test for the GLMM). Covariates and interactions were stepwise removed if non-significant[89] and reported p-values refer to the final model without non-significant covariates or interactions. An overview of all full models can be found in Supplementary Table 13. The residuals and Q/Q plots of all LMMs were visually inspected, and the distributions of the residuals were compared to a normal distribution using Kolmogorov–Smirnov and Shapiro tests. If the residuals were non-normally distributed, a log transformation was applied and the residuals again checked. The GLMM was checked for overdispersion, but did not require a correction.

To control for differences between blocks we always included 'block ID' as a random effect. If individual data for both subjects from the same trial were used, we also included 'sister dyad ID' as a random effect. To analyse whether outgroup competition influenced the response variables we used a factor 'Outgroup competition' with three levels to test for an overall effect of outgroup competition across both experiments, and for separate effects of unrelated competitors within the same territory or as neighbours in an adjacent territory. To analyse whether the number of protected nest sites available influenced the response variables we used a factor 'Protected nest sites' with two levels: multiple or single protected nest sites available. We checked for any remaining variation explained by differences between experiments using the factor 'experiment' with two levels (experiment 1 or 2). However, this factor was not a significant predictor of the response variable in any of the analyses and was dropped from the final models.

Reproductive success was quantified as the number of weaned offspring produced. Data on individual (but not combined) reproductive success could not be analysed for two sister dyads where maternity of weaned offspring could not be fully discriminated. In one trial, illness-related late-stage offspring mortality occurred prior to collection of samples for maternity analysis, and data on both individual and combined reproductive success were excluded from analysis for the subjects in this trial (for more details on the sample sizes used for each analysis see Supplementary Table 1). To test if the relative time spent in the nest by communal breeding subjects was influenced by their average PVN oxytocin concentrations, we calculated the difference in the proportion of total time spent in the nest by each subject (subtracting the lower from the higher value), separated into active (dark: 0800-20.00) and inactive (light: 20.00-0800) periods. The maximum sample size for this analysis was constrained by the number of cases in which PVN oxytocin data were available for both subjects sharing a communal nest ($n = 15$ sister dyads). To test if the reproductive skew of communal breeding sisters was related to their average PVN oxytocin concentrations, we calculated the difference in the number of offspring weaned by each subject within a sister dyad, and controlled for the combined number of weaned offspring produced by both subjects.

### Reporting summary

Further information on research design is available in the Nature Portfolio Reporting Summary linked to this article.

## Data availability

The experimental datasets that support the findings of this study are available in Figshare with the identifier https://figshare.com/s/c4c981f8a7cdc7153c0d[90].

## Code availability

The R-code used to generate all models and figures presented in this study is available in Figshare with the identifier https://figshare.com/s/c4c981f8a7cdc7153c0d[90].

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

## Acknowledgements
We thank John Waters, Richard Humphries and Simon Gibson for technical support and animal care. Manuela Ferrari and members of the Mammalian Behaviour and Evolution Group provided helpful discussions. We are also grateful to three reviewers for constructive comments. This study was funded by a research grant to P.S. & J.L.H. from the Natural Environment Research Council (NERC, NE/M002977) and by NERC funded PhD studentships to C.D. and R.L.B. supervised by P.S. and J.L.H.

## Author contributions
Conceptualization/Funding acquisition/Supervision: P.S. & J.L.H.; Experimental design: P.S., J.L.H., S.F., C.D. & W.T.S.; Methods development: S.F., C.D., J.L.H., R.L.B., W.T.S. & A.J.D.; Data collection: C.D., S.F., A.J.D. & W.T.S.; Data curation: C.D. & S.F.; Data analysis: S.F. & P.S. with input from W.T.S. & C.D.; Writing – original draft: P.S. & S.F. with input from all authors; Writing – review & editing: all authors.

## Competing interests
The authors declare no competing interests

## Ethics
All animal care protocols were in accordance with the University of Liverpool Animal Welfare Committee requirements and UK Home Office guidelines for animal care. Tissue samples for maternity analysis were obtained under UK Home Office licence according to the best practice guidelines and approved by the University of Liverpool Animal Welfare Committee. We have complied with all relevant ethical regulations for animal use.
