## [Transparent Peer Review file · Communications Biology]

Egalitarian cooperation linked to central oxytocin levels in communal breeding house mice

Corresponding Author: Professor Paula Stockley

Version 0:

Reviewer comments:

Reviewer #1

(Remarks to the Author)

This is a well written, interesting and important study and for most sections of the paper I have only minor clarification points to raise. However, I have some methods concerns that need addressing prior to publication of the study, concerning the modelling of potential confounding variables, how model fit was assessed and clarifying several points on the experimental setup and validation of the ELISA used to detect the oxytocin in the PVN of the study mice. Finally, while the topic concerned and the paper's findings are of broad interest the authors need to better develop these points in the discussion section. Please see below for specific feedback on each section.

Specific comments:

Abstract

Very well written although rather light on methods description.

Introduction

Overall very well written, I have some minor clarification comments;

L47 – surely this is relevant for understanding co-operation in all animals, vertebrate or invertebrate? Not sure why the invertebrates have been excluded here.

L50 – true that philopatry provides an opportunity for this but as dispersal is common in vertebrate reproduction need to provide some quantification of how common this is in this statement.

L59-60 – while the rest of this paragraph is strong, the final sentence is rather weak and non-specific. Why is this needed? Or what specifically is needed that ties to a need to study neuropeptide dynamics?

L79-80 – 'remains largely unexplored', too ambiguous, this has either been studied in the past, in which case you should briefly state what was found here, or has not been studied, in which case just outright state it.

L95 – don't need 'also' in this line

L98-99 – "A commensal existence means that food is not usually limiting for wild house mice under natural conditions."

Why? Need to explain this for those not familiar with house mice.

L101-102 – why are you assuming plasticity here? Can you provide some evidence that this is reasonable to expect?

L118-119 – can you provide a statement on the appropriateness of using levels in the PVN as a way to measure oxytocin dynamics, e.g. by citing previous studies that have done so too successfully? If the references at the end of the statement do this, you need to explicitly state this in the sentence.

Figure 1 legend – while I like this figure, the part of the legend explaining what all the various sizes/colours/symbols etc within it should come earlier in the legend, currently you have to read a lot of text without knowing what means what on the figure to get to that information.

Results

Main take home results are written clearly however there are some confusing points and some information missing, which may be due to the reading order of this journal but still needs to be accounted for in the manuscript;

L131-131 – the sample size numbers are hard to follow, this needs clarifying. Figure 1 shows four mice in experiment A setups and four to six in experiment B setups, so how can samples be taken from 'both' subjects when there is always more than 2 individuals per trial, or which are the study individuals within these groups? Or are the numbers in figure 1 not accurate and are just a representation? What constitutes a trial in this project? The number of mice used in what trials and which were sampled needs to be much better explained. These details may be present in the methods section below, but as the results section comes first in this journal this needs to be clearer at this point in the manuscript.

L143-152 – need to state the statistical test used along with the summary statement of the test here, and the legend for the supplementary tables should also state what sort of model/analysis is being done (applies to all table legends and results text but I won't repeat myself for every instance of this happening).

L178-179 – ‘control for combined litter size’ do you mean ‘controlling for combined litter size? Also, unclear how you have done this, hopefully it is explained in the methods section.

Discussion

Concise and well written but several points need more development and the ties to broader importance needs to be improved, currently there is very little discussing this;

L212 – “with higher levels mediating more successful and egalitarian cooperation between sisters.” Need to add ‘when breeding communally’

L227-229 – need to state what species/strain this was studied in (from paper 10), house mice, lab strains etc

L258-263 – while I agree that measuring basal levels is important and your inference about it predicting oxytocin release is fine, you need to explore what the functional relevance of elevated oxytocin is in this brain region and what could be driving it, social cues from bonded individuals for example? Is there data on non-breeding individual levels in the PVN to compare to, either from your own work or other publications?

L286-288 – yes I agree and think this point needs to be better developed. You cite supporting studies here, how plausible is the evidence supporting this hypothesis? What sensory cues or behaviours that are specific to the sisters rather than the pups could be driving this?

L293-294 – should mention though that there is a decent amount of papers showing oxytocin is linked to outgroup responses, so it is interesting that you didn’t see anything here and potentially worth following up on, especially as you controlled aggressive responses for ethical reasons. I would expect behavioural changes like you mention, but not necessarily changes to basal levels in the PVN, were any of these monitored?

L306-309 – Need to discuss the broader literature more here, mentioning the one vampire bat study is not enough. For example, there are several chimpanzee papers on co-operation and oxytocin dynamics, specifically one on food sharing, that should also be mentioned.

L311-316 – conclusions and end of the discussion needs to be stronger, need to link the findings to the broader significance of the results.

Methods

The methods text clarifies some points but I have concerns that need addressing. Some statements cause confusion and doubt about the experimental setup and how the various values tested were calculated, this must be addressed prior to publication. Also, no justification or control in the models for the large range of time (7-19 days) the females were left prior to sampling after pups were weaned. Finally, some more detail needed from the validation/quality control aspect of running the ELISA plates;

L336-337 – Here is the detail about who in the trial was considered the subject, this needs to be in the results section for this reading order.

L337 – surely all mice were given the opportunity to raise their offspring communally, not just the older sisters?

L344-345 – this implies that younger sisters were not mated and could not breed, but how could they then produce pups and thus enable them to breed communally with their sisters? And how could you find that “sisters’ average PVN oxytocin concentrations were positively related to the total number of weaned offspring produced when a communal nest was formed” (L161-162) if only one sister was breeding? This is now very confusing, needs to be clarified who was breeding and how you calculated the variables you tested if only one female was breeding in the pair of sisters forming the ‘communal’ nest. Surely the nest cannot be communal if only one of them is breeding?

L354-356 – “maternity analysis of weaned offspring to confirm that both sisters had contributed to a communal litter”, this statement implies that both sisters were mated at some point, which makes sense in terms of how the experiment works but contradicts what is stated in L344-345. Needs fixing.

L377-379 – why were the individuals not immediately sacrificed and PVN oxytocin levels measured? Need to justify this choice, the range in time left after pup removal is also large, 7-29 days? What evidence do you have that this induced no variation in the PVN levels detected? Even if you have evidence, this should be included as an explanatory variable in your statistical analysis to rule this out, and I’m hoping you did this.

Supplementary methods 1, L123-124 – what test did you use to analyse parallelism? Without knowing what test you used, the summary statement is not informative of whether you were successful, in my statistical testing method for parallelism a significant result, which you report, indicated interaction between lines and therefore no parallelism, so this needs some more explanation. Also, was no recovery tests for the validation? This is a standard part of the process, and you did spike samples so calculating this should be straightforward?

L396-397 – provide mean and ranges for CoV values for inter and intra assay variation, and how many plates were run in total.

L409 – no mention of how model fit was assessed, this needs to be stated. Non-significant terms can and should be retained if they improve fit but you state all were removed here?

L411-421 – clear explanations, but no mention of ‘number of days until sample collection post-weaning’. This should be included in the initial models to rule this out as a serious confounding factor. Also, no mention of when linear or GLM approaches were used, and what error distribution was used when using GLM.

Ethical statement

L642 – need to provide home office licence number and state whether individuals who carry an appropriate personal licence carried out all procedures?

Reviewer #2

(Remarks to the Author)

Fischer et al. conducted an interesting study concerning the relationship between brain oxytocin (OT) concentrations and maternal behavior (including its consequences to the offspring) of communally breeding female sister dyads. The paper is generally well written and structured. The experiments were well conducted and the data was analyzed thoroughly.

The authors show three main results:

- 1) OT is related to reproductive success in dyads but not in single breeders.
- 2) OT is related to reproductive skew in dyads.
- 3) OT is related to time spent in nest in dyads.

These results suggest that OT coordinates maternal care of both sisters in a dyad, resulting in a better outcome for the offspring. The authors also provide an interesting negative result, namely that OT does not seem to be linked to the propensity of becoming a communal breeder.

However, there are some issues about the presented data and one issue about the methods.

The first issue is about point 2) of the above-mentioned main results. The authors show that OT is related to reproductive skew in dyads, however, they don't show any data for single breeders. Why is this result not shown similarly as in 1)? The same applies to point 3). The authors show that OT is related to time spent in nest in sister dyads but they don't compare this to the data of the single breeders.

Lastly, it did not become clear in the methods sections how exactly the pups have been tracked. There is a statement about RFID chips but it is not written if the pups or the mothers were implanted with them and if the pups were implanted, when did this happen? It would be important to describe how the pup-to-mother identity could be tracked "unambiguously" from birth, as the authors describe in their first results paragraph ("252 of 256 weaned offspring").

Additionally, it may be good to address the potential limitation of this study such as generalizability of the findings to other populations or species.

Another point is the absence of an effect of individual PVN oxytocin concentrations on the number of weaned offspring, therefore, it will be important to discuss potential explanations for this lack of association.

Taken together, I suggest that the manuscript should be published in *Communications Biology* if the above-mentioned concerns can be addressed.

Reviewer #3

(Remarks to the Author)

In this study, the authors test for relationships between PVN OT levels of female mice and the reproductive/parental behavior of these mice. The authors house animals in relatively large enclosures that enable them to identify if females communally or singly rear their offspring.

The authors find that PVN OT levels are similar between sisters in dyads, and that communally rearing sisters with higher PVN OT concentrations had increased reproductive success and less variance in reproductive outcomes (number of offspring weaned). The authors also find that females that have lower asymmetry in nest attendance had more similar PVN OT levels.

The study is well-designed and the data analysis are sound. The paper is very well written and the rationale for conducting the experiment is well founded. The data and code are presented and analyses are able to be reproduced.

I just have a few clarifying comments:

1- Can the authors confirm that all females were nulliparous at the beginning of the experiment?

2- For models where residuals were not normal, did the authors attempt to fit other distributions to the model (e.g. gamma, exponential) before log transforming data - it is possible that they would fit the data better.

3- Table S2. It might be valuable to include this correlation as a supplemental scatterplot also.

4- Post-weaning, subjects were left for between 7-29 days prior to decap for oxytocin assessment. It is noted that timing of sample collection is used as a random effect in models. However, I think it is worth describing if PVN OT levels are affected directly by the timing of decap. There is strong reason to expect that oxytocin levels may decline gradually following weaning.

5- I find how some of the statistical models are discussed in terms of prediction to be a little counter-intuitive. For example, "Average PVN oxytocin concentrations of sister dyads did not predict whether or not they both bred and formed a communal nest". I understand what the authors are describing here - they performed a mixed-effect model using OT concentration as a fixed factor predictor. However, it is important that the oxytocin was measured some time after the behavior. In that sense, biologically it feels that the behavior is just as (or more) likely to hypothetically induce changes in the PVN OT levels, as the other way around. I do not object to use of the word 'predict' here in the statistical sense, but I think some readers may be confused. Perhaps just some extra discussion that the proposed relationships between these variables may biologically work in both directions.

I think a related point to this is Table S10. Here, Average PVN OT is listed as an independent variable. Often we think of these variables as those that we experimentally manipulate such that we can observe changes in the dependent variables. However, here the authors do not manipulate PVN OT levels, but as using the observed data to infer associations with dependent variables. I would just urge caution in how describing causality here. I do appreciate the paragraph in the discussion that deals with this issue.

6- I understand the rationale for examining if average OT levels of sister dyads was related to the relative time that each spent in the nest. However, it would also seem logical to present data examining how absolute levels of nest time are related to individual OT levels. It would seem that with such a dataset as this, that this would be worth testing and reporting.

7- I think the correlation between sister dyads in OT concentrations is very notable. I also thought that this was evaluated nicely in the discussion. I think it would worth this being in the main manuscript as opposed to a supplemental figure.

Version 1:

Reviewer comments:

Reviewer #2

(Remarks to the Author)

The authors performed comprehensive revision and addressed all my concerns. I recommend acceptance of this interesting study for publication.

I have also carefully read authors' responses to points raised by reviewer 1. I think that all points were properly addressed. Thus I suggest to publish this very interesting work.

Reviewer #3

(Remarks to the Author)

I thank the authors for their considerable attention to reviewers' comments. The authors have satisfactorily responded to each of my major concerns. I think that this paper will be a welcome addition to the literature.

Authors' response to reviewers' comments

We are very grateful to the reviewers for their helpful comments, attention to which has greatly improved the manuscript. We have made significant revisions throughout in addressing these comments. Most notably these include the addition of new data and analyses as requested to strengthen our conclusions. We have added further explanation throughout the manuscript to clarify potential misunderstandings, as well as providing more details concerning our statistical approach, methodology and assay validations. We have also added a new figure to the main text (moved from SI) as requested. These and further revisions are explained in more detail below. We hope that you will agree that the revision is much improved.

Reviewers' comments:

Reviewer #1 (Remarks to the Author):

This is a well written, interesting and important study and for most sections of the paper I have only minor clarification points to raise. However, I have some methods concerns that need addressing prior to publication of the study, concerning the modelling of potential confounding variables, how model fit was assessed and clarifying several points on the experimental setup and validation of the ELISA used to detect the oxytocin in the PVN of the study mice. Finally, while the topic concerned and the paper's findings are of broad interest the authors need to better develop these points in the discussion section. Please see below for specific feedback on each section.

Specific comments:

Abstract

1. Very well written although rather light on methods description.

We appreciate this comment. With respect to methods, we are constrained by the abstract word limit so have focused on the key concepts and findings rather than methodological details.

Introduction

Overall very well written, I have some minor clarification comments;

2. L47 – surely this is relevant for understanding co-operation in all animals, vertebrate or invertebrate? Not sure why the invertebrates have been excluded here.

We have deleted 'of vertebrate animals' in the first sentence of the introduction, and also from the abstract and the end of the discussion.

3. L50 – true that philopatry provides an opportunity for this but as dispersal is common in vertebrate reproduction need to provide some quantification of how common this is in this statement.

We have changed this sentence to: '*Female relationships are of particular interest in mammals, since female philopatry is common and provides opportunities for kin selected benefits of cooperation*', and added a reference for female philopatry – Greenwood 1980 (L51).

4. L59-60 – while the rest of this paragraph is strong, the final sentence is rather weak and non-specific. Why is this needed? Or what specifically is needed that ties to a need to study neuropeptide dynamics?

We have added an additional sentence to the end of this paragraph (L61-63):

A focus on the underlying neural mechanisms is of particular relevance to understanding patterns of social behaviour and cooperation associated with kin selection.

5. L79-80 – ‘remains largely unexplored’, too ambiguous, this has either been studied in the past, in which case you should briefly state what was found here, or has not been studied, in which case just outright state it.

Okay, thanks. We have deleted ‘largely’ – now says ‘remains unexplored.’

6. L95 – don’t need ‘also’ in this line

We would prefer to keep this in, to improve flow from the previous paragraph / make the connection with the point about mice being a model species for related (but different) questions.

7. L98-99 – “A commensal existence means that food is not usually limiting for wild house mice under natural conditions.” Why? Need to explain this for those not familiar with house mice.

We have added (L103-104): *‘as typically they live in close proximity to abundant food supplies in agricultural or domestic settings.’*

8. L101-102 – why are you assuming plasticity here? Can you provide some evidence that this is reasonable to expect?

We are not so much assuming plasticity as testing for it – this was a nuance of the wording. We have reframed this, and now say (L108): *‘Hence if plasticity exists in central oxytocin production ..’*, citing Fricker *et al.* (2023) which suggests that PVN nonapeptide cell groups are particularly plastic in adulthood.

9. L118-119 – can you provide a statement on the appropriateness of using levels in the PVN as a way to measure oxytocin dynamics, e.g. by citing previous studies that have done so too successfully? If the references at the end of the statement do this, you need to explicitly state this in the sentence.

The cited references here (51,52) support the statement that the PVN is the main source of centrally released oxytocin. We address the reviewer’s point as to whether other studies have used a similar approach to ours in the discussion (L288-291), where we explain that the approach of quantifying basal oxytocin expression is relatively unusual but that a relationship between basal oxytocin expression in the PVN and social behaviour has previously been demonstrated in rats [ref 27]. This study quantified oxytocin based on mRNA expression. Baseline PVN oxytocin levels have also been quantified previously using an ELISA method by Tait *et al.* (2009) [ref 71]. This reference is also cited in the discussion (L305).

10. Figure 1 legend – while I like this figure, the part of the legend explaining what all the various sizes/colours/symbols etc within it should come earlier in the legend, currently you have to read a lot of text without knowing what means what on the figure to get to that information.

We have moved the text explaining the symbols higher up in the legend as requested.

Results

Main take home results are written clearly however there are some confusing points and some information

missing, which may be due to the reading order of this journal but still needs to be accounted for in the manuscript;

11. L131-131 – the sample size numbers are hard to follow, this needs clarifying. Figure 1 shows four mice in experiment A setups and four to six in experiment B setups, so how can samples be taken from ‘both’ subjects when there is always more than 2 individuals per trial, or which are the study individuals within these groups? Or are the numbers in figure 1 not accurate and are just a representation? What constitutes a trial in this project? The number of mice used in what trials and which were sampled needs to be much better explained. These details may be present in the methods section below, but as the results section comes first in this journal this needs to be clearer at this point in the manuscript.

The older sister pairs within each group are the subjects in each trial. This is explained in the legend to Figure 1 (L781-803) and again in the methods (L388-395). This design allows us to compare how subjects respond across different social environments, while minimising variation in the number of females breeding at any one time, which would impact relative investment in communal litters. The experimental set-up also reflects typical breeding patterns in natural populations, where older females are dominant and tend to monopolise breeding opportunities.

To further emphasise this point, we have added additional text at the beginning of the results section (L139-148):

Each experiment consisted of a series of independent trials (16 for experiment 1 and 20 for experiment 2 – see Table S1), in which the social environment of female house mice was manipulated (Fig 1). Subjects within each trial were a pair of littermate sisters living with two younger non-breeding females. This design replicates a naturally occurring social structure, where older females tend to monopolise breeding opportunities^{45,46}. Experiment 1 manipulated the availability of protected nest sites and the relatedness of younger competitors living within the subjects’ territory, while experiment 2 manipulated the presence or absence of unrelated competitors in a neighbouring territory (Fig. 1). The experiments thus included subjects experiencing outgroup competition with unrelated females within their own territory or in a neighbouring territory.

We have also reworded the methods section to further emphasise this point (L392-394):

In experiment 1, the environment was manipulated by varying the availability of protected nest sites, and the relatedness of younger females living within the subjects’ territory (two sisters or an unrelated sister dyad of equivalent age).

We have also made minor adjustments in wording throughout the text when referring to subjects and sisters, which will hopefully further minimise the potential for confusion.

12. L143-152 – need to state the statistical test used along with the summary statement of the test here, and the legend for the supplementary tables should also state what sort of model/analysis is being done (applies to all table legends and results text but I won’t repeat myself for every instance of this happening).

We now clearly explain in the table legends in the main manuscript and supplementary information which model has been used. In the methods section (L468-469) we have added a statement to clarify how we obtained p-values from linear mixed models (LMM) and generalized linear mixed models (GLMM). This statement should make it clear to readers that whenever we present F-values we refer to results from a LMM whereas when we refer to χ^2 -values we refer to results from a GLMM.

13. L178-179 – ‘control for combined litter size’ do you mean ‘controlling for combined litter size? Also, unclear how you have done this, hopefully it is explained in the methods section.

We have changed the wording here to ‘controlling for combined litter size’ (L195 & 199). This approach is explained in the methods (L500-503).

Discussion

Concise and well written but several points need more development and the ties to broader importance needs to be improved, currently there is very little discussing this;

14. L212 – “with higher levels mediating more successful and egalitarian cooperation between sisters.” Need to add ‘when breeding communally’

Done (L235).

15. L227-229 – need to state what species/strain this was studied in (from paper 10), house mice, lab strains etc

All the studies cited here refer to house mice (not laboratory mice), as specified in line 247-248: ‘*Although oxytocin has not been demonstrated to influence social bonds between female house mice ..., there is evidence ...*’

Elsewhere (e.g. L235-237) we point out where studies use laboratory mice, and why wild-derived house mice are more suitable subjects for studying social relationships (L235-242).

16. L258-263 – while I agree that measuring basal levels is important and your inference about it predicting oxytocin release is fine, you need to explore what the functional relevance of elevated oxytocin is in this brain region and what could be driving it, social cues from bonded individuals for example?

We’re not sure what additional exploration the reviewer is looking for here, so apologies if we’re missing the point. From our perspective, exploring the functional relevance of variation in basal PVN oxytocin levels is already a key theme throughout the manuscript. That is, we’ve shown that there is variation in this trait, and we’ve shown it is linked to cooperative behaviour and reproductive success. However, we haven’t yet been able to explain how the variation arises, as there was no evidence of plasticity in response to the social cues that we manipulated – and this forms a key part of the discussion.

As noted by reviewer 3, given that no evidence of plasticity in PVN oxytocin production was found in response to manipulation of the broader social environment, a potentially important finding is the correlation in values we found for cohabiting sisters (following reviewer 3’s suggestion we have now moved Fig S1 to the main text). Hence we discuss whether this correlation may be linked to variation in the strength of social relationships between sisters, perhaps influenced by social feedback mechanisms, and/or whether basal PVN oxytocin levels may be a heritable trait, perhaps reflecting differences in oxytocin gene expression, or in the number or size of oxytocin neurons, as has been shown with species differences in cooperative behaviour (Reddon *et al.* 2017 [74]; L309-316). We have further extended this discussion point in response to point 17 (see below).

Is there data on non-breeding individual levels in the PVN to compare to, either from your own work or other publications?

We have already included a comparison of females that bred versus those that didn't (L169-170), highlighted in bold): *Similarly, PVN oxytocin concentrations were not significantly influenced by subjects' age or body mass (Table S3), **whether they successfully reared offspring or not** (Table S4 factor 'Whether subject produced weaned offspring' [$F_{1,47.12}=0.03$, $p=0.85$])*

We also have unpublished data from a different study, outlined below (in response to point 17), using non-breeding females.

17. L286-288 – yes I agree and think this point needs to be better developed. You cite supporting studies here, how plausible is the evidence supporting this hypothesis? What sensory cues or behaviours that are specific to the sisters rather than the pups could be driving this?

We think the hypothesis is plausible based on evidence from previous studies. In the discussion (L247-259), we explain how female house mice form consistent social preferences prior to breeding. These typically involve sleeping with preferred partners in shared nests. Importantly, previous studies also show that when females breed communally with a *preferred partner* they are more likely to form egalitarian cooperative relationships, resulting in higher reproductive success. So, if the strength of female relationships is linked to their PVN oxytocin levels, this could explain our results. Currently, we can't say that a social feedback mechanism is the most likely explanation for correlated PVN oxytocin levels within sister dyads in our study, because sisters may have similar PVN oxytocin levels due to being genetically similar and/or sharing the same early life environment. Importantly though, unrelated females can also form social preferences for one another (Weidt *et al.* 2008 [11]) and show egalitarian cooperation during communal breeding (Green *et al.* 2023 [53]). Hence further experiments could be designed to tease apart the extent to which variation in relatedness and early-life experience versus the strength of relationships formed explains the correlation in PVN oxytocin levels that we report here for sister dyads.

We have added further text to the discussion (L318-324):

Consistent with this idea, previous evidence suggests that female house mice are more likely to show egalitarian cooperation if they form an affiliative social relationship prior to breeding¹¹. Although such relationships are more usually formed between kin, both affiliative social relationships and egalitarian cooperation can also occur between unrelated females^{11,53}. Hence there is potential for future studies to tease apart the effects of relatedness and strength of social relationships in explaining the association between hypothalamic oxytocin levels and egalitarian cooperation reported here.

In a different experiment (currently in prep.), we have found evidence that basal PVN oxytocin levels predict the strength of social relationships among related female house mice prior to breeding. We found that non-breeding females with higher PVN oxytocin levels chose to spend more time close to a familiar sister, but there was no relationship between PVN oxytocin levels and time spent close to an unfamiliar sister (in both cases, subjects were offered a choice of spending time near to a sister or an unfamiliar unrelated female). These results support the idea that co-habiting sisters with higher PVN oxytocin levels have stronger social relationships prior to breeding. Again however, the data don't reveal the mechanism by which variation in PVN oxytocin levels arises (e.g. via social feedback mechanisms, shared early life experience and/or genetic differences between lineages). Further work is needed to explore this.

18. L293-294 – should mention though that there is a decent amount of papers showing oxytocin is linked to outgroup responses, so it is interesting that you didn't see anything here and potentially worth following up on, especially as you controlled aggressive responses for ethical reasons.

We have reworded the beginning of this paragraph to remind readers of papers cited earlier in the manuscript, showing that oxytocin is linked to outgroup responses (L326-327):

Despite growing evidence linking oxytocin to outgroup responses^{13,25}, we found no evidence of plasticity in oxytocin production linked to outgroup competition.

We have also added a further suggestion to the discussion, as we will soon have new data to test this idea (L333-336):

Alternatively, plasticity in basal oxytocin expression in response to out-group competition may be dependent on aspects of subjects' phenotype that were not quantified in the current study. For example, responsiveness to an out-group threat might vary according to a subject's competitive behaviour or dominance status⁸⁰.

I would expect behavioural changes like you mention, but not necessarily changes to basal levels in the PVN, were any of these monitored?

We monitored scent marking behaviour of subjects throughout the study to confirm that expected behavioural responses to out-group competitors did occur (as has been demonstrated in the previous studies cited). However, as competitive scent marking behaviour isn't the main focus of the present study, and our study is already quite complex to explain, we plan to report these findings separately.

An additional analysis now included (explained in the section of this document headed 'Other changes and corrections' below) does reveal a behavioural response to neighbours, although this is unrelated to PVN oxytocin levels:

L217-221: Average oxytocin levels did not predict relative time in the nest during inactive (light) periods (Table S10 factor: "Average PVN oxytocin" [$F_{1,8}=0.16$, $p=0.7$]), although a significantly greater skew in time spent in the nest during inactive (light) periods was associated with the presence of outgroup competitors in the neighbouring territory (Table S10, S11).

L326-330: Despite growing evidence linking oxytocin to outgroup responses^{13,25}, we found no evidence of plasticity in oxytocin production linked to outgroup competition. Nonetheless, it is unlikely that subjects in our study were unresponsive to competitors. For example, we found that the presence of neighbours influenced relative time spent in the nest during periods when the mice are typically resting.

19. L306-309 – Need to discuss the broader literature more here, mentioning the one vampire bat study is not enough. For example, there are several chimpanzee papers on co-operation and oxytocin dynamics, specifically one on food sharing, that should also be mentioned.

We have added a new general sentence at the beginning of this paragraph, to remind readers of earlier cited examples linking cooperation and oxytocin dynamics in wild mammal populations (L343-344):

Oxytocin has previously been associated with cooperative behaviour in wild mammals, including examples of alloparental care³⁴, social grooming³⁵, food sharing³⁶, and cooperative defence¹³.

We have also cited the vampire bat study at the end of this paragraph as a specific example where oxytocin influenced the degree of cooperation with a given partner, rather than the propensity to cooperate. This links directly to the previous point in the text, in relation to our findings.

20. L311-316 – conclusions and end of the discussion needs to be stronger, need to link the findings to the broader significance of the results.

We have added new text to further emphasise and explain the broader significance of the results (L358-368):

Social competition and conflict shape the social systems of group living animals, with diverse outcomes influenced by kin selection and benefits of cooperation^{81,82}. The resulting tension between competition and cooperation is reflected by variation in how benefits of cooperative behaviour are distributed between group members. In egalitarian social systems, benefits are shared relatively evenly according to effort invested, whereas in despotic social systems, benefits are more likely to accrue disproportionately to dominant individuals at the expense of others^{4,83}. Our study provides evidence of variation in the balance between egalitarian and despotic outcomes linked to central oxytocin levels of cooperating individuals. If similar variation is replicated across species, this could help us to understand the proximate factors influencing egalitarian and despotic social behaviours, hence providing broad insight to social system diversity.

Methods

The methods text clarifies some points but I have concerns that need addressing. Some statements cause confusion and doubt about the experimental setup and how the various values tested were calculated, this must be addressed prior to publication. Also, no justification or control in the models for the large range of time (7-19 days) the females were left prior to sampling after pups were weaned. Finally, some more detail needed from the validation/quality control aspect of running the ELISA plates;

21. L336-337 – Here is the detail about who in the trial was considered the subject, this needs to be in the results section for this reading order.

As outlined in response to point 11, we have added a new paragraph at the beginning of the results section, so that the general approach is clear without having to read ahead to the methods (L130-146):

Each experiment consisted of a series of independent trials (16 for experiment 1 and 20 for experiment 2 – see Table S1), in which the social environment of female house mice was manipulated (Fig 1). Subjects within each trial were a pair of littermate sisters living with two younger non-breeding females. This design replicates a naturally occurring social structure, where older females tend to monopolise breeding opportunities^{45,46}. Experiment 1 manipulated the availability of protected nest sites and the relatedness of younger competitors living within the subjects' territory, while experiment 2 manipulated the presence or absence of unrelated competitors in a neighbouring territory (Fig. 1).

22. L337 – surely all mice were given the opportunity to raise their offspring communally, not just the older sisters?

No, as stated, the trials were standardised by only allowing the two older sisters to breed, so that we could focus on their relationships and reproductive success (in each trial the subjects are a pair of litter-mate sisters – the two older females in the group). We now emphasise more strongly that this represents a naturally occurring social structure, similar to natural populations where younger females within a social group are less likely to be breeding (L142-143 – see response to point 21). The younger females in each group are potential competitors to the breeding females.

23. L344-345 – this implies that younger sisters were not mated and could not breed, but how could they then produce pups and thus enable them to breed communally with their sisters? And how could you find that “sisters’ average PVN oxytocin concentrations were positively related to the total number of weaned

offspring produced when a communal nest was formed” (L161-162) if only one sister was breeding? This is now very confusing, needs to be clarified who was breeding and how you calculated the variables you tested if only one female was breeding in the pair of sisters forming the ‘communal’ nest. Surely the nest cannot be communal if only one of them is breeding?

To clarify - each social group consists of four females – two pairs of sisters (Fig 1). The older females come from the same litter as one another (they are litter-mates – the same age - sister dyads) and are the subject of the experiments. They may choose to breed communally with one another, or not. The younger females also come from the same litter as one another (but different to the older females, obviously, as they are not the same age). They can be related to the older sisters or not (this was part of the manipulation in experiment 1) – i.e. they may have the same or different parents to the older females. The younger females did not mate or produce offspring. Hence they did not form communal nests with the older females.

We have adjusted the legend to Fig 1 and hope the design is now clearer. Please also note that in the figure, females that share the same parents (and hence are full sisters) are represented in the same colour.

24. L354-356 – “maternity analysis of weaned offspring to confirm that both sisters had contributed to a communal litter”, this statement implies that both sisters were mated at some point, which makes sense in terms of how the experiment works but contradicts what is stated in L344-345. Needs fixing.

We appreciate that the reviewer is finding the description of sisters confusing, and perhaps is expecting a breeding sister pair should consist of one older and one younger female, rather than two older females, as the study is designed. Please see response to point 23.

25. L377-379 – why were the individuals not immediately sacrificed and PVN oxytocin levels measured? Need to justify this choice, the range in time left after pup removal is also large, 7-29 days? What evidence do you have that this induced no variation in the PVN levels detected? Even if you have evidence, this should be included as an explanatory variable in your statistical analysis to rule this out, and I’m hoping you did this.

The delay between lactation and measurement of PVN oxytocin levels was a deliberate part of the experimental design, because the questions that we are aiming to address relate to the environment experienced throughout the study (i.e. the relationship between the adult females that are the subjects in each experiment, and the presence or absence of outgroup competition as well as the availability of protected nest sites for breeding). If we had recorded PVN oxytocin levels of subjects immediately when their offspring were weaned, the results may have been influenced by the subjects’ recent lactation, which might mask the ‘background’ patterns we were looking for. Hence we waited a minimum of 7 days (actually 8 days was the shortest, now corrected) after removing weaned offspring at age 28-30 days (now explained L434). Taking into account that lactation ends by around 23 days, this gives a median of 21 days post-lactation before females were culled to record oxytocin levels. If lactation influenced the results despite this delay, we would have expected to see a difference in the average PVN oxytocin values of females that had reared a litter versus those that didn’t. As reported in the results (L169-170, Table S4), no such difference was found.

Beyond the minimum delay period built into our study, variation in the time between lactation and measurement of oxytocin levels arose mainly due to timing the end of the trials to coincide with the availability of trained animal care staff to assist with Schedule 1 procedures. We were relatively flexible in these timings, since as long as the subjects were being maintained consistently under the same experimental conditions, the precise timing of when the trial ended shouldn’t affect the predicted outcomes. Nonetheless,

we agree it is important to check directly if there was any effect of variation in the time since lactation on PVN oxytocin levels and have added a new analysis to confirm this (L171-172, Table S5).

We have also added new text to explain that we left time for a post-breeding recovery period before quantifying oxytocin levels (L130-132):

Experiments allowed sufficient time for social groups to become established, for subjects to complete a reproductive cycle, and for a post-breeding recovery period prior to quantifying oxytocin levels.

And to emphasise that we deliberately avoided recording oxytocin levels during the period of cooperation when females were lactating (L293-295):

Since our aim was to allow subjects to exhibit natural behaviour over a complete breeding cycle, we did not attempt to quantify central oxytocin release during periods of cooperation, when lactation may also have influenced results.

26. Supplementary methods 1, L123-124 – what test did you use to analyse parallelism? Without knowing what test you used, the summary statement is not informative of whether you were successful, in my statistical testing method for parallelism a significant result, which you report, indicated interaction between lines and therefore no parallelism, so this needs some more explanation.

There seems to be some variation in the literature with respect to how this is reported. We appreciate the reviewer's suggestion here and agree their approach makes more sense. We have added a new analysis to show there is no significant interaction between lines and therefore successful parallelism was achieved (SI L122-124):

Serial dilutions of pooled PVN samples resulted in a displacement curve parallel to that of the standard curve (no difference in the slopes following linear regression, $F = 0.006$, $P = 0.94$, $n = 7$).

27. Also, was no recovery tests for the validation? This is a standard part of the process, and you did spike samples so calculating this should be straightforward?

Yes, we have added this information to the SI (L124-128):

To conduct a matrix interference assessment, serial dilutions of synthetic oxytocin standard (125, 250, 500 and 1000 pg/ml) were spiked with equal volumes of sample at a working dilution of 1:8. No significant interference was found, as confirmed by linear regression analysis ($R^2 = 0.99$, $F = 203$, $p = 0.005$, $n=4$). Average recovery was 93.4%.

28. L396-397 – provide mean and ranges for CoV values for inter and intra assay variation, and how many plates were run in total.

We have added this information to the methods (L454-458):

All samples were run in duplicate alongside a standard curve (in triplicate) on a total of five plates. In addition to the in-plate controls supplied by the manufacturer, two further controls were used in duplicate on each plate at 25% (control 1) and 75% (control 2) binding. Intra-assay CVs were less than 10% for all samples included in the analyses (average 2.6%), and inter-assay CVs were less than 15% (average 13.4%).

29. L409 – no mention of how model fit was assessed, this needs to be stated. Non-significant terms can and should be retained if they improve fit but you state all were removed here?

Model fit of all linear mixed effect models was assessed by visual inspection of Q/Q plots and histograms of residuals. We applied log transformations in cases where residuals and Q/Q plots showed skewness in the data. Model fit for the generalized linear mixed model was checked by assessing overdispersion. However, no correction as needed. We have now added this information in the methods section (L471-475).

We did not simply remove all non-significant terms from the model but instead followed the approach of Engvist 2005 [89] to remove non-significant covariates and interactions. Variables were classified as explanatory variables or as covariates *a priori* depending on the main focus of the analysis (see Table S13) and explanatory variables were not removed even if not significant. For example, when testing whether PVN oxytocin levels vary in response to the experimental manipulations we included the factors 'protected nest site', and 'outgroup competition' as predictors without removing them if not significant (see Table 1).

30. L411-421 – clear explanations, but no mention of 'number of days until sample collection post-weaning'. This should be included in the initial models to rule this out as a serious confounding factor.

See response to point 25 and additional analysis provided (L171-172, Table S5).

31. Also, no mention of when linear or GLM approaches were used, and what error distribution was used when using GLM.

This information has now been added throughout the manuscript.

Ethical statement

32. L642 – need to provide home office licence number and state whether individuals who carry an appropriate personal licence carried out all procedures?

The ethical statement in our manuscript is very similar to the one published in our recent 2023 *Communications Biology* paper, so we assume meets the requirements for the journal unless this has recently changed. We do not normally cite specific Home Office licence numbers as licences in the UK are not published so the number has no external use (only anonymous summaries are publicly available) but we can include this if it is journal policy. As work under UK Home Office licence can only be carried out by trained individuals holding a suitable current Personal licence, working under an awarded Project licence, at an Institution holding an Establishment licence and within specific locations covered under the Establishment licence, stating that the work was carried out under Home Office licence confirms that individuals carrying out procedures held Personal licences to do so. Please do let us know if the journal requires further information added to the ethics statement.

Reviewer #2 (Remarks to the Author):

Fischer et al. conducted an interesting study concerning the relationship between brain oxytocin (OT) concentrations and maternal behavior (including its consequences to the offspring) of communally breeding female sister dyads. The paper is generally well written and structured. The experiments were well conducted and the data was analyzed thoroughly.

The authors show three main results:

- 1) OT is related to reproductive success in dyads but not in single breeders.
- 2) OT is related to reproductive skew in dyads.
- 3) OT is related to time spent in nest in dyads.

These results suggest that OT coordinates maternal care of both sisters in a dyad, resulting in a better outcome for the offspring. The authors also provide an interesting negative result, namely that OT does not seem to be linked to the propensity of becoming a communal breeder. However, there are some issues about the presented data and one issue about the methods.

1. The first issue is about point 2) of the above-mentioned main results. The authors show that OT is related to reproductive skew in dyads, however, they don't show any data for single breeders. Why is this result not shown similarly as in 1)?

The short answer is that there would be no variation in reproductive skew for single breeders, since, by definition, reproductive skew = 1 when only one female in a pair has any reproductive success.

However, we can also expand our response to explain conceptually why we focus our analysis of reproductive skew on communal breeding females, and why this is relevant to interpreting Fig. 3 (to which the reviewer is referring as result 1).

The reason for looking at reproductive skew is to explore the nature of the social relationship between communally breeding sisters. As explained in the introduction, communal breeding involves a risk of exploitation, with potential for unequal investment in the shared litter. If communally breeding females have a despotic relationship, then a more dominant female may reduce the number offspring produced by her partner, and thereby gain a greater share of investment for her own offspring (e.g. by inhibiting the fertility of the other female). Alternatively, if communally breeding females have a more egalitarian relationship, their investment and reproductive success is likely to be shared more equally. Our findings indicate that when sister dyads have relatively high basal PVN OT levels, their relationship is more likely to be an egalitarian one.

Our analysis of reproductive skew complements and extends the results shown in Fig 3. As explained in the manuscript (L55-57; 93-98), our ultimate goal here is to explore the extent to which the relationship between PVN OT and reproductive success is explained by variation in the social relationship between cooperating subjects. If the social relationship between the subjects is important, we expected to see evidence of a relationship between the OT levels of communally breeding females and their reproductive skew. We conducted further analyses also to check if the quality of maternal care might be a confounding factor explaining variation in the number of weaned offspring produced, but we found no evidence to support this (183-189). See also our response to point 5, where we have added new text to further explain this.

2. The same applies to point 3). The authors show that OT is related to time spent in nest in sister dyads but they don't compare this to the data of the single breeders.

Again, this is because we are investigating the nature of the relationship between subjects that are breeding communally – i.e. those that are showing cooperative behaviour. If only one subject within a pair is lactating,

there is no cooperative behaviour between them (shared nursing of the pups), and hence no reason to compare the relative time being spent in the nest by each subject.

However, in response to a comment by reviewer 3, we have added a new analysis to test if PVN OT levels predict absolute time spent in the nest by all subjects, rather than just relative time spent in the nest by communally breeding subjects (L221-223) – see also below in response to point 5.

3. Lastly, it did not become clear in the methods sections how exactly the pups have been tracked. There is a statement about RFID chips but it is not written if the pups or the mothers were implanted with them and if the pups were implanted, when did this happen? It would be important to describe how the pup-to-mother identity could be tracked “unambiguously” from birth, as the authors describe in their first results paragraph (“252 of 256 weaned offspring”).

Thanks for pointing this out. We identified maternity of pups using microsatellite analysis (lines L408-410 and Supplementary methods S1). Pups were not tagged (this would not be a practical way to determine maternity as females often give birth on the same day, plus the pups are too small to be tagged at birth and we wouldn't want to disturb the nest). We have adjusted the methods to specify that RFID tags were used for individual identification of adult mice (L384).

4. Additionally, it may be good to address the potential limitation of this study such as generalizability of the findings to other populations or species.

We think that the findings should be fairly generalizable to other species, not least because we have used wild-derived mice with normal behaviour and genetic variation as subjects rather than laboratory strains. Communal breeding is a relatively unusual behaviour in mammals but the social bonds and cooperative behaviour required to co-ordinate this behaviour is typical of many social animals.

We have added new text to the discussion, explaining how our findings may be of broad significance if generalizable to other species (see response to reviewer 1 point 20).

Elsewhere we point out where potential limitations of our study occur, for example with respect to managing social interactions to avoid direct aggression (L336-341):

Moreover, although our study was designed to imitate natural conditions, subjects were not free-living and, for ethical reasons, we managed social interactions between unrelated competitors to avoid a risk of escalated aggression. Hence, although subjects were continuously exposed to social odours of competitors, reinforced with controlled physical contact, we cannot rule out that their response to competitors may be different in unconstrained natural populations.

5. Another point is the absence of an effect of individual PVN oxytocin concentrations on the number of weaned offspring, therefore, it will be important to discuss potential explanations for this lack of association.

We appreciate this suggestion. We have added new text to the discussion to further address this point, and to consider the new analysis provided to test for a relationship between oxytocin levels and absolute time spent in the nest (L261-272):

We found no evidence that PVN oxytocin concentrations explained variation in reproductive success of females breeding alone, or of individual females. Absolute time spent in the nest with pups was also unrelated to PVN oxytocin levels. Hence, we found no evidence that relatively high PVN oxytocin levels were associated with more or better-quality maternal care per se. Although studies of laboratory mice confirm that oxytocin has an essential role in milk ejection, such studies also report normal levels of maternal

behaviour in oxytocin deficient females⁶¹⁻⁶³, including those with conditional knock-out of oxytocin in the PVN⁶⁴. Hence our finding that natural variation in PVN oxytocin levels of wild-derived house mice has no apparent influence on their weaned offspring numbers or maternal care behaviour in the absence of communal breeding is not unexpected. Rather, our findings suggest that the PVN oxytocin levels of sister dyads reflect variation in the strength of their social relationships, with consequences for the combined reproductive success and reproductive skew of sisters that cooperate to breed communally.

Taken together, I suggest that the manuscript should be published in Communications Biology if the above-mentioned concerns can be addressed.

Reviewer #3 (Remarks to the Author):

In this study, the authors test for relationships between PVN OT levels of female mice and the reproductive/parental behavior of these mice. The authors house animals in relatively large enclosures that enable them to identify if females communally or singly rear their offspring.

The authors find that PVN OT levels are similar between sisters in dyads, and that communally rearing sisters with higher PVN OT concentrations had increased reproductive success and less variance in reproductive outcomes (number of offspring weaned). The authors also find that females that have lower asymmetry in nest attendance had more similar PVN OT levels.

The study is well-designed and the data analysis are sound. The paper is very well written and the rationale for conducting the experiment is well founded. The data and code are presented and analyses are able to be reproduced.

I just have a few clarifying comments:

1- Can the authors confirm that all females were nulliparous at the beginning of the experiment?

Yes. Now confirmed in the methods (L383).

2- For models where residuals were not normal, did the authors attempt to fit other distributions to the model (e.g. gamma, exponential) before log transforming data - it is possible that they would fit the data better.

In short, we did not attempt to fit other distributions to the models because in all cases the log transformation provided a good fit. We are aware of the discussion about whether or not data should be transformed to meet the requirements of a linear approach. Although there are benefits in using other distributions of generalized linear models we used a parsimonious approach and followed recommendations in Ives (2014) to avoid inflated type I errors.

*Ives, A.R. (2014) *Methods in Ecology & Evolution*: doi/10.1111/2041-210X.12386*

3- Table S2. It might be valuable to include this correlation as a supplemental scatterplot also.

This was originally provided as Fig S1. In response to the suggestion made in point 7 we have now moved this figure to the main text as Fig 2.

4- Post-weaning, subjects were left for between 7-29 days prior to decap for oxytocin assessment. It is noted that timing of sample collection is used as a random effect in models. However, I think it is worth describing if PVN OT levels are affected directly by the timing of decap. There is strong reason to expect that oxytocin levels may decline gradually following weaning.

Thanks yes, we should have included this as an extra analysis. This is now added, and confirms there is no effect of the time between weaning date and culling on PVN oxytocin levels. See response to reviewer 1 point 25 and additional analysis provided (L171-172, Table S5).

5- I find how some of the statistical models are discussed in terms of prediction to be a little counter-intuitive. For example, "Average PVN oxytocin concentrations of sister dyads did not predict whether or not they both bred and formed a communal nest". I understand what the authors are describing here - they performed a mixed-effect model using OT concentration as a fixed factor predictor. However, it is important that the oxytocin was measured some time after the behavior. In that sense, biologically it feels that the behavior is just as (or more) likely to hypothetically induce changes in the PVN OT levels, as the other way around. I do not object to use of the word 'predict' here in the statistical sense, but I think some readers may be confused. Perhaps just some extra discussion that the proposed relationships between these variables may biologically work in both directions.

We appreciate this point. It was something that we debated in preparing the manuscript, and why we make a point of emphasising in the introduction that we are looking at OT measured after the behaviour was recorded.

As suggested, we have added a further caveat to the discussion (L284-285): *No direct causal link between oxytocin and behaviour is demonstrated by our results, and we are unable to confirm directionality in reported relationships between oxytocin levels and behaviour.*

Elsewhere in the discussion we also point out the possibility of a feedback mechanism between social interactions and oxytocin levels, such that socially rewarding behaviour influences oxytocin production and *vice versa* (L316-318).

I think a related point to this is Table S10. Here, Average PVN OT is listed as an independent variable. Often we think of these variables as those that we experimentally manipulate such that we can observe changes in the dependent variables. However, here the authors do not manipulate PVN OT levels, but as using the observed data to infer associations with dependent variables. I would just urge caution in how describing causality here. I do appreciate the paragraph in the discussion that deals with this issue.

Thank you, we had missed this. We have now changed the terms used in Table S10 (now Table S13) to explanatory variable (instead of independent) and response variable (instead of dependent). These seem more appropriate. We have also added further comment on the uncertainty of causality to the discussion (see previous point).

6- I understand the rationale for examining if average OT levels of sister dyads was related to the relative time that each spent in the nest. However, it would also seem logical to present data examining how

absolute levels of nest time are related to individual OT levels. It would seem that with such a dataset as this, that this would be worth testing and reporting.

We agree with this suggestion. In response we have added new analyses to test if absolute levels of time in the nest with pups are related to individual OT levels. This is now included as Table S12, and mentioned in the results and discussion (L221-223; 261-272; see also reviewer 2 point 5).

7- I think the correlation between sister dyads in OT concentrations is very notable. I also thought that this was evaluated nicely in the discussion. I think it would be worth this being in the main manuscript as opposed to a supplemental figure.

Done. Now Fig 2 in the main manuscript.

Other changes and corrections:

In addition to the revisions described above in relation to the reviewers' comments we have made several further minor corrections and additions to the manuscript. None of these have altered the main findings or conclusions of the study.

1. We should have specified that the results in Fig. 4 (the relationship between skew in time spent in the nest by communally breeding sisters and their average PVN oxytocin concentrations) relate to the period when the mice are typically active (i.e. the dark period - in our animal unit this is when lights are off between 0800-2000). We now explain this, and for completeness we have added a complementary analysis to the SI for the period when the mice are typically resting (i.e. the light period - in our animal unit this when lights are on between 2000-0800). This shows a response to the presence of neighbours in the relative time spent resting (L218-221).

2. In collating the data for absolute time spent in the nest, as requested by reviewer 3, we identified an error in the code used to collect these results (put simply, some recording durations had overrun, which was more obvious when dealing with absolute time in the nest rather than relative proportions of time). Correcting this, and a minor inconsistency in the way that results were calculated for experiments 1 and 2, has altered some data points in our analysis of relative time spent in the nest (although not dramatically so, as the proportion of total time spent in the nest tends to stay relatively constant when measured over a longer duration). Following this correction we were able to increase the number of data points that can be included in the analysis, since access to multiple nest boxes is no longer having a significant effect on relative time spent in the communal nest. The new Fig. 4 and associated results in Table 3 thus appear slightly different in the revision.

3. We have corrected the range of days between removing weaned offspring and collecting brain samples (L437: 8-34 days).

4. We have rearranged the order of variable names and definitions in Table S13 (previously S12) to match the revised order of models and to include new variable names resulting from the additional analyses provided.

5. Further small changes in wording have been made throughout the manuscript to improve clarity of explanation with respect to subjects / sisters, that caused some confusion to the referees.

We have also updated our datafiles and R-code used to generate all models and figures contained in the manuscript, and uploaded these to the Figshare repository, accessible via the following link: <https://figshare.com/s/c4c981f8a7cdc7153c0d>.